# Blockchain-Based Traceability Architecture for Mapping Object-Related Supply Chain Events

**DOI:** 10.3390/s23031410

**Published:** 2023-01-27

**Authors:** Fabian Dietrich, Louis Louw, Daniel Palm

**Affiliations:** 1Department of Industrial Engineering, Stellenbosch University, 145 Banghoek Rd., Stellenbosch 7600, South Africa; 2ESB Business School, Reutlingen University, Alteburgstr. 150, 72762 Reutlingen, Germany; 3Fraunhofer Institute for Manufacturing Engineering and Automation, Alteburgstr. 150, 72762 Reutlingen, Germany

**Keywords:** blockchain, tokenisation, object traceability, EPCIS events

## Abstract

Supply chains have evolved into dynamic, interconnected supply networks, which increases the complexity of achieving end-to-end traceability of object flows and their experienced events. With its capability of ensuring a secure, transparent, and immutable environment without relying on a trusted third party, the emerging blockchain technology shows strong potential to enable end-to-end traceability in such complex multitiered supply networks. This paper aims to overcome the limitations of existing blockchain-based traceability architectures regarding their object-related event mapping ability, which involves mapping the creation and deletion of objects, their aggregation and disaggregation, transformation, and transaction, in one holistic architecture. Therefore, this paper proposes a novel ‘blueprint-based’ token concept, which allows clients to group tokens into different types, where tokens of the same type are non-fungible. Furthermore, blueprints can include minting conditions, which, for example, are necessary when mapping assembly processes. In addition, the token concept contains logic for reflecting all conducted object-related events in an integrated token history. Finally, for validation purposes, this article implements the architecture’s components in code and proves its applicability based on the Ethereum blockchain. As a result, the proposed blockchain-based traceability architecture covers all object-related supply chain events and proves its general-purpose end-to-end traceability capabilities of object flows.

## 1. Introduction

Due to globalisation, supply chains have evolved from traditional linear supply chains to static supply networks and have further evolved into complex interconnected supply networks [1]. Such interconnected networks can exhibit complex buyer–supplier relationships with both parties involved simultaneously, even competing in supply chains [1]. Furthermore, in contemporary interconnected supply chains, most products travel in a non-predefined manner through a supply chain network, resulting in supply chains that can emerge unpredictably over time [2].

In addition to the structural complexities of contemporary supply chains, recent trends increasingly put pressure on companies to increase their supply chain visibility and to provide supply chain transparency to maintain their competitiveness [3]. Here, the literature often uses the terms ‘supply chain visibility’ and ‘transparency’ interchangeably. However, Barratt and Oke define supply chain visibility as “the extent to which actors *within* [emphasis added] a supply chain have access to or share information which they consider as key or useful to their operations and which they consider will be of mutual benefit” [4]. In comparison, Sodhi and Tang define supply chain transparency as the extent to which supply chain actors disclose information to *all* stakeholders, including the public, consumers, and investors [3]. Thus, supply chain visibility enables companies to provide supply chain transparency [2].

For companies, traceability represents the essential prerequisite for enabling supply chain visibility [3,5], which in turn represents the prerequisite for providing supply chain transparency [2]. The term traceability and the use of traceability systems originated in food supply chains and have spread from there to various industries [6,7,8]. Olsen and Borit define traceability “as the ability to access any or all information relating to that which is under consideration, throughout its entire life cycle, by means of recorded identifications” [7]. According to the GS1 Global Traceability Standard, traceability information can relate to the origin of materials and parts, processing history, and distribution and location. Interconnected traceability systems map objects through their object-related supply chain events [2], also referred to as object-related ‘visibility events’ [9]. Such event-based mapping approaches belong to the discrete mapping domain, capturing event data in discrete milestones [10]. The counterpart of discrete mapping approaches is continuous mapping, which collects data at predefined time intervals. However, compared to the more widespread discrete mapping approaches, continuous mapping is used mainly in specific tracking solutions, such as continuous humidity sensor measurements in food supply chains [10].

“At the heart of any traceability system is the identification of traceable objects” [2]. The international standard IEC 62507 specifies basic requirements for systems to identify objects [11]. Here, the standard distinguishes between physical and abstract objects. The GS1 Global Traceability Standard shares this distinction between objects, except that it refers to abstract objects as digital objects [9]. Based on these two international standards, this paper uses the following definition of objects: *Physical objects*. Refers to physical objects “which are handled in physical handling steps of an overall business process involving one or more organisations” [9]. This includes objects such as products, items, and physical documents and explicitly excludes human individuals [11].*Abstract objects*. Refers to digital objects which participate in business processes involving one or more organisations. This includes objects such as digital trade items, digital documents, and electronic certificates [9].

Information systems must map objects in their information world to enable object traceability [12]. According to the IEC 62507 standard, the identification number refers to an object or a group of objects (for example, in assemblies consisting of several components [13]) and links to the metadata. The metadata contains all the relevant information to describe an object or a group of objects. On a system level, metadata includes data components, which “may be simple real numbers, text strings, vectors of real numbers and other values, sets in real vector spaces, functions from real vector spaces to other data spaces, or complex combinations of these” [14]. Furthermore, traceability systems typically connect objects’ virtual representations via radio frequency identification (RFID) tags or quick response (QR) codes to their physical counterparts [15]. 

The traceability of objects requires mapping data related to the occurred supply chain events [2], also referred to as object-related ‘visibility events’ [9]. An object-related supply chain event “is the record of the completion of a specific business process step acting upon one or more objects” [9]. Here, the Electronic Product Code Information Services (EPCIS) standard—which represents the most frequently applied standard in industrial traceability systems [10]—defines the following core supply chain events: Object creation and deletion, object aggregation and disaggregation, object transformation, and object transaction events [9]: *Object event* [9]. Object events initially link objects to their identifiers (IDs) on a system level and include simple observations of objects identified in the event. In addition, object events allow the possibility to create a number of objects of the same object class. In addition to the creation of objects, object events can delete objects, which results in these objects not existing on a system level for further events after their deletion.*Aggregation/disaggregation event* [9]. Aggregation events create a new identifiable ‘containing’ entity that contains a set of objects. Until their possible disaggregation, aggregated objects, physically and on a system level, occupy the same location at the same time. Consequently, aggregation events include the possibility of disaggregating previously aggregated objects, whereas the ‘containing’ objects become independent objects again, and the ‘containing’ entity dissolves.*Transformation event* [9]. Transformation events fully or partially consume objects as inputs and produce outputs of new object classes. Like this, objects can ‘transform’ into new objects without experiencing changes regarding their modular composition.*Transaction event* [9]. Transaction events associate or disassociate objects with business transactions. Therefore, transaction events enable the mapping of objects’ changes in ownership.

The increasing requirements regarding a higher need for supply chain visibility and transparency have led to the rapidly growing efforts of companies to map out their supply chains [16]. Nevertheless, it is still common practice that companies only have visibility of their direct suppliers and customers—also known as the ‘one step up-one step down model’ [2]—eventually leading to overall limited knowledge of their supply chains [16]. 

Alternatively, companies can rely on traditional centralised systems to ensure traceability throughout their supply chains, which bundle the traceability data across all parties in a central repository [2]. However, globalised interconnected supply chains, with each party having a particular interest in increasing the supply chain visibility and providing transparency, cannot rely on focal companies enforcing a system for their whole supply chain. Therefore, as the first discussions of the German Federal Ministry for Economic Affairs and Energy with various companies of different industry sectors and regions imply, an exclusive commitment to a central system is challenging to achieve [17]. 

Consequently, companies proceeded with cumulative traceability approaches, where each party stores the traceability data centrally but pushes them to the next party in parallel with the product flow [2]. Such a cumulative traceability approach does not require an exclusive commitment to a superordinate central system; however, it results in novel challenges, particularly for downstream parties, when receiving and processing large traceability data volumes [2]. To overcome these challenges, the ‘Plattform Industrie 4.0′ initiative summarises several international standardisation approaches to define a common standard for the information exchange of objects between partners in value chains, also known as Asset Administration Shell (AAS) [18]. Here, the AAS represents a standardised digital representation of an asset aiming to facilitate interoperability when pushing asset-related data from one company’s system to another company’s system [18]. 

The advantage of the AAS not currently requiring a superordinate central system also results in its weaknesses in a traceability context. Since no central system monitors the parties’ compliance with the standardised formats, evaluations of the approach show that applying the AAS increases the semantics’ complexity, leading to data inconsistencies [19,20]. Furthermore, the problem of data inconsistencies compounds due to the fact that no central entity ensures the global uniqueness of identifiers, applying to both the identification of objects and of experienced services or events [18]. Therefore, in large and complex supply chains, these data inconsistencies eventually result in big data problems instead of facilitating object traceability [20]. 

In 2016, Abeyratne and Monfared published the first concept adopting an emerging technology—blockchain technology—in manufacturing supply chains as a potential solution for solving their end-to-end traceability problems [21]. Blockchain technology, which was first introduced by the pseudonym Satoshi Nakamoto in 2008 [22], nowadays represents the most famous representative of the distributed ledger technology (DLT) family [23]. A DLT describes a “multi-party system in which all participants reach consensus over a set of shared data and its validity in the absence of a central coordinator” [24]. Here, the primary difference between blockchain and other forms of DLT relates to the storage of data [23,25]. “In a blockchain, data is stored as groups, or ‘blocks’, of information. New transactions can only add information to the ‘chain’ of past transactions; it is impossible to delete or modify information previously stored ‘on the chain’ because blocks are replicated across multiple ledgers” [25]. Taking into account the development of blockchain technology with the various consensus algorithms available [26], this paper extends the DLT definition of Hansen [24] with blockchain-specific characteristics and provides the following definition of blockchain technology:


*Blockchain technology is a multi-party system in which all participants or an agreed fraction of participants reach a consensus over shared transaction data summarised in linked data blocks and their validity, resulting in a linear and immutable chain of data blocks without requiring a central coordinator.*


In particular, blockchain technology’s capabilities of providing a superordinate system without requiring a trusted third party while still ensuring a secure, transparent, and immutable environment with globally unique ‘digital profiles’ brought it onto the map as a potential solution to the problem of achieving end-to-end traceability in complex multitiered supply networks [21,27,28]. As a review and bibliometric analysis conducted by Fang, Fang, Hu, and Wan reveal, over the years, technology-wise, the term ‘blockchain’ has become the most frequently used keyword related to supply chain management [29]. Accordingly, various blockchain-based traceability solutions have arisen, making blockchain technology arguably the “most promising technology for providing traceability-related services in supply chain [abbreviation deleted] networks” [30]. However, the current blockchain-based solution landscape implies that each traceability problem requires an individual architecture design and blockchain platform [30]. As exploitation of blockchain-based traceability solutions indicates, most architectures, particularly the food supply chain and medical supply chain, show simple architecture designs that can only map single objects throughout the supply chain and lack the ability to cover all supply chain events defined by EPCIS [31].

In considering the requirements for dynamic, interconnected supply networks and the previously mentioned limitations of current solutions to ensure end-to-end traceability, the following problem arises: To the best of the authors’ knowledge, no general-purpose blockchain-based architecture for dynamic, interconnected supply networks exists to ensure the traceability of object flows and their experienced supply chain events: creation and deletion, aggregation and disaggregation, transformation, and transaction.

Therefore, the following primary research question (PRQ) aims to solve the identified problem by developing a blockchain-based traceability architecture:*PRQ* *How can a blockchain-based traceability architecture be constructed which meets the general-purpose requirements of dynamic, interconnected supply networks and ensures the end-to-end traceability of object-related supply chain events?*

In addition, the following secondary research questions (SRQ) support answering the PRQ and prove the architecture’s applicability and completeness.

*SRQ1* 
*What are the limitations of existing blockchain-based traceability solutions described in the literature?*
*SRQ2* 
* What are the architectural requirements for an end-to-end traceability solution for dynamic, interconnected supply networks?*
*SRQ3* 
*How can the architecture’s components be implemented in code to enable its practical applicability in a blockchain-based traceability solution?*


This paper aims to propose a novel blockchain-based traceability architecture to overcome the limitations of existing solutions. For this purpose, Section 2 first describes the research methodology adopted in this study. After this, Section 3 analyses existing advanced blockchain-based traceability architectures regarding limitations in terms of their mapping abilities. Following this, the architecture development takes place in Section 4. Subsequently, Section 5 evaluates the proposed architecture based on a prototype to assess its practical applicability. Finally, this paper concludes the research results by answering the research questions, pointing out the unique contributions, summarising the architecture’s current limitations, and suggesting recommendations for further research.

## 2. Research Methodology

This paper aims to develop a novel blockchain-based traceability architecture. The innovativeness of such design calls for conducting the research according to the design science research strategy. “Design science research involves the creation of new knowledge through the design of novel or innovative artifacts (things or processes) and analysis of the use and/or performance of such artifacts along with reflection and abstraction—to improve and understand the behavior of aspects of Information Systems” [32]. In this paper, the blockchain-based traceability architecture represents the artefact to be developed. According to the design science research strategy, the development of such an artefact incorporates five process steps: Awareness of the problem, suggestion, development, evaluation, and conclusion [32]. 

Section 1 raises the awareness of traceability problems in contemporary supply chains with emergent characteristics based on recent publications and an initial exploration of the literature. Here, the study extends the methodology of Vaishnavi and Kuechler [32] by constructing design science research questions according to Hoang Thuan, Drechsler, and Antunes [33], aiming to support solving the initial problem statement.

Section 3 provides a theoretical knowledge foundation regarding related works in the supply chain and blockchain technology domain and defines essential terms. Furthermore, this section emphasises the research gap to ensure the paper’s uniqueness by pointing out the limitations of existing blockchain-based traceability architectures. 

In Section 4, the development of the architecture takes place. Here, an architecture describes “the set of structures needed to reason about the system, which comprise software elements, relations among them, and properties of both” [34]. This does not include its implantation in code; however, its design and logic can lead to code that is correct with respect to the specified architecture [35]. For the development processes, this paper adopts parts of the architecture development framework of Vogel et al. [36]. As is common in software development, architecture development begins with deriving architectural requirements [37]. Requirements “provide the key input to the software architecture design” [38]. This study derives the architectural requirements from the fundamental supply chain structures, the characteristics of traceability systems, and the available object-related traceability standards. According to the process flow of the design science research strategy, this paper suggests an initial architecture design based on the derived requirements and subsequently works out the artefact’s detailed structure and components. 

Following the architecture development, this research evaluates the developed architecture based on a prototype. Prototyping-based evaluation methods belong to the most commonly used techniques in the industry when evaluating novel software architectural designs and may uncover aspects that other evaluation methods are not able to identify [39]. Here, according to Gregor and Hevner’s design science research theory, this paper aims to develop an ‘instantiated artefact’ representing a fully operational software [40]. Therefore, this paper evaluates the applicability of all architecture components in code and applies a prototypical implementation in an experimental setting to ensure their implementation ability. 

Subsequent to the prototyping-based evaluation, the paper concludes the research results and points out their unique contribution by answering the research questions and stating the key findings. Finally, the architecture’s current limitations are summarised and recommendations are suggested for further research.

## 3. Limitations of Existing Blockchain-Based Traceability Architectures

A systematic review of the literature by Chang and Chen [41] reviewed potential blockchain applications and their development status in the supply chain management domain to identify future trends. The authors identify that the vast majority of blockchain-based publications address traceability and transparency issues in supply chains [41]. A further review by Dasaklis et al. [30] focuses on the implementation state of blockchain-enabled traceability solutions. The authors point out that even though blockchain-enabled traceability implementations encompass various supply chain domains, they currently lack advanced and functional interfaces and validations in industrial settings, making it difficult to assess the quality of the proposed solutions [30]. Furthermore, an initial exploration of blockchain-based traceability solutions implies that most solutions only incorporate architectures of low complexity with the ability to map single objects without compositional changes [42]. However, these low-complexity architectures have already proven to increase supply chain visibility and effectively reduce operational costs in industrial settings. For example, during a 12-month trial, the blockchain-based container tracking solution ‘TradeLens’ by Maersk and IBM showed a reduction in the shipment transit time by 40% and demonstrated the potential for some supply chain partners to reduce the efforts required to locate a container from ten steps and five people to one step and one person [43].

When further investigating the traceability solutions identified in the systematic literature reviews regarding their ability to map supply chains of high complexity with objects experiencing compositional changes, the literature dominantly references three advanced traceability architectures by Westerkamp et al. [44], Watanabe et al. [45], and Kuhn et al. [46]. These advanced traceability architectures utilise the tokenisation of objects and provide token ecosystems that allow users to conduct object-related supply chain events in an arbitrary sequence. In addition, these solutions describe logic to aggregate tokens with individual token IDs and ‘merge’ them into a new token, which allows on-chain mapping of compositional changes, for example, when assembling components.

In general, blockchain tokens are “blockchain-based abstractions that can be owned and that represent assets, currency, or access rights” [47]. Since tokens can reflect various states of their representatives and interact with each other, these token-based blockchain architectures allow linking object-related supply chain event data providing traceable production information along the supply chain [44]. 

Figure 1 shows a functionality overview of the most relevant ERC (Ethereum request for comments) token standards. The figure adapts the general structure and illustration format of Wang et al. [48] by correcting the relationships according to the current specifications of the ERC-20, ERC-721, and ERC-1155 token standards [49,50,51]. The following paragraph summarises the token standards’ key characteristics: *ERC-20.* The ERC-20 represents the first fungible token (FT) standard. Its functions initially define the total token supply and provide a simple logic to transfer tokens from one address to another [49].*ERC-721.* The ERC-721 represents the first non-fungible token (NFT) standard. Its functions initially create a token with a unique identifier and provide a simple logic to transfer the unique tokens from one address to another [50].*ERC-1155.* The ERC-1155 represents the first multi-token standard. Its functions initially define NFT types, which allow the creation of a group of FTs of the same type and provide a simple logic to transfer NFTs and FTs from one address to another [51]. If only one token of a particular type exists, it shows similar characteristics to the NFTs of the ERC-721 [51].

Taking into account the object-related supply chain events specified by EPCIS and the capabilities of the applied token standards, an investigation of the advanced traceability architectures reveals the following limitations when being applied in dynamic, interconnected supply chains involving complex objects with the ability to experience compositional changes: *Governance concept*. The architecture proposed by Kuhn et al. [46] represents the only architecture incorporating a governance concept managing the parties involved. However, the governance is part of the selected blockchain platform, with the traceability architecture exclusive to the complete blockchain. Therefore, it is not possible to transfer the governance concept’s logic to blockchain platforms that are not exclusive to the architecture and require the administration of parties involved on an application level.*Token deletion*. The advanced architectures do not describe the possibility of an explicit token deletion without requiring a functional workaround. For example, the traceability architectures by Westerkamp et al. [44] and Watanabe et al. [45] provide logic for ‘consuming’ tokens. Here, consumed tokens receive a mark indicating their state to avoid the reusability of consumed tokens in further token recipes. Kuhn et al. [46] describe a similar logic but refer to the consumption of tokens as the ‘burning’ of tokens. Even though the logic to consume or burn tokens intentionally serves as functionality to avoid the reusability of tokens, for example, after assembling processes, this logic also allows the creation of a token recipe to remove tokens from the supply chain. Although none of these three architectures further describe this procedure, a recipe that consumes or burns its input tokens supposedly results in a new, albeit useless, ‘waste token’. Therefore, strictly speaking, this logic does not allow the deletion of tokens in the sense of EPCIS [9].*Token aggregation*. Kuhn et al. [46] point out the ill-suited capabilities of the ERC-721 NFT standard when mapping objects with great variety and assembly complexity. As a solution, Kuhn et al. [46] adopt the ERC-1155 token standard; however, this only allows minting FT batches of the same type. Therefore, the ERC-1155 solves the problem of the ERC-721 when applying it to batches of fungible assemblies of various fungible components, such as those incorporated by the electrical and electronic system case study of Kuhn et al. [46]. However, when mapping multiple non-fungible assemblies of the same type with non-fungible inputs of the same type, the ERC-1155 results in the same limitations as the ERC-721.*Token disaggregation*. Among the advanced architectures, only the architecture developed by Watanabe et al. [45] describes a mechanism for token ‘forking’. The architectures of Westerkamp et al. [44] and Kuhn et al. [46] merely include a logic for ‘splitting’ token batches, which describes distributing a share of a token batch to different owners. Westerkamp et al. [44] even view the absence of an ability for token disaggregations to be a limitation of their architecture and refer to a possible example of packaging processes, which require the extraction of the original good when unpacking [44]. Even though the ‘forking’ described by Watanabe et al. [45] forks a token into two tokens, these forked tokens receive new identifiers and new smart contract addresses, which does not ‘restore’ the previously aggregated tokens and, therefore, does not solve the limitation mentioned by Westerkamp et al. [44], representing a disaggregation according to EPCIS [9].

## 4. Architecture Development 

The paper’s architecture uses elements of the general structure of decentralised applications (dApps) as a fundamental frame for the proposed architecture. This includes the interplay of interfaces, smart contracts, and the underlying blockchain. 

Even though the literature often uses the terms ‘smart contract’ and ‘dApp’ interchangeably, the fundamental difference is that dApps have a user interface while smart contracts and tokens do not [52]. Smart contracts in a blockchain context represent software scripts deployed on the blockchain [53], while “dApps within a dApp ecosystem comprise a user interface and one or more smart contracts which interact with a blockchain [emphasis deleted]” [52]. As is common for blockchain-based traceability architectures, linking physical objects to their token representatives relies on RFID tags or QR codes and readers as supporting infrastructure [44,46]. Since the architecture intends to be a dApp that uses the underlying blockchain only as an operating system, involved parties require only a blockchain account (typically managed by a wallet provider) and must not provide server infrastructure to run blockchain nodes. 

The following section initially derives the architectural requirements. Subsequently, the development of the governance and token concept takes place. 

### 4.1. Requirement Derivation

This paper derives the architectural requirements from fundamental supply chain structures, characteristics of traceability systems, available object-related traceability standards, and limitations of available advanced blockchain-based traceability architectures. The following list enumerates the architecture’s fundamental requirements (R):*R1*. The general structure of traceability systems consisting of participating parties and objects at their heart provided by the GS1 Global Traceability Standard [2] requires the architecture to map this fundamental structure.*R2*. Each party has a specific role in the value-adding process [54] and must be identifiable to ensure trust [2], which requires the architecture to identify each party and provide a clear assignment of rights. *R3*. Interconnected supply chains can experience structural transactions at any time [1] and require the architecture to allow certain parties a dynamic adding and removing of parties. *R4*. Objects in traceability systems must be identifiable [11] and can experience creations and deletions, aggregations and disaggregations, transformations, and transactions [9], which require the architecture to identify each object and enable the mapping of their object-related events. *R5*. In emergent supply chains, objects travel in a non-predefined manner through supply chains [2], which requires the architecture to allow objects to experience events in arbitrary sequences.*R6*. Companies must gain supply chain visibility and offer supply chain transparency.throughout the entire supply chain [3], which requires the architecture to ensure a traceable event history throughout objects’ entire life cycles.

Figure 2 shows the initial architecture design suggestion based on the derived architectural requirements. As shown, the initial design includes a governance concept for mapping the supply chain structure with a token concept embedded therein for mapping object-related supply chain events.

### 4.2. Development of a Governance Concept 

The governance concept develops functions defining the supply chain structure-related administrative capabilities of the dApp and manages all parties registered to the dApp in a party memory. The functions of the governance concept include possibilities to add and remove parties as well as to edit their structure-related administrative and object-related operative rights.

#### 4.2.1. Adding Parties

According to the architectural requirements, it is essential for the responsible authority (or authorities) to ensure the deanonymisation of each party involved and remove parties with negative behaviour from the application so as not to violate the overall integrity of the traceability dApp. Therefore, the architecture’s smart contract requires a governance concept, which defines the corresponding logic for adding and removing partiers. Here, the study refers to the function set, summarising supply chain structure-related functions as the *governance set*. Additionally, the governance set includes a data memory managing all parties registered to the dApp—the *party memory*. Before deploying the governance set, the code must contain or define an initial account administrator with the ability to add further accounts to the party memory. In this architecture, as typical for many dApps, the deploying account becomes the initial administrator. 

The GS1 Global Traceability Standard states that a traceability system requires linking each party that plays a role in the chain of custody or ownership of a supply chain to a unique identifier and storing crucial party-related data [2]. Typically, this contains data such as contact data and data describing the respective supply chain’s respective roles. With its accounts consisting of a public and private key, blockchain technology already ensures the uniqueness of accounts and the traceability of each account’s blockchain interaction. However, as Kuhn et al. describe, it is not only necessary to link each public key to data defining a supply chain party but also to display it understandably [46]. 

Here, dApps offer two possible solutions: Either an off-chain storage stores the data and the interface links them with the corresponding public key or the smart contract stores the data on-chain. Since the deanonymisation of supply chain parties represents a crucial requirement of the traceability dApp, the study’s architecture suggests storing crucial party-related data on-chain in the party memory of the governance set. On the one hand, data such as contact data and role descriptions are relatively small in storage size. On the other hand, this ensures an immutable link between public keys and party-related information, regardless of the interface used.

#### 4.2.2. Removing Parties

Removing data entries from the blockchain refers to the general smart contract logic described by Hu et al. [55]. Accordingly, removing a party updates the governance set and removes the respective party from the party memory. This change affects all blocks after the ‘removal-transaction’ confirmation. However, while removed parties no longer have access to the dApp’s functions in current blocks, deprecated blocks still prove their past participation. 

#### 4.2.3. Editing Rights

Each party has a specific role in the supply chain’s value-adding process, which requires a clear assignment of rights. Smart contracts generally allow a functional coupling of any function to certain requirements. The study’s architecture proposes a rights distinction according to its general structure to reduce the number of functional dependencies. Therefore, the architecture distinguishes between administrative supply chain structure-related rights and operative object-related rights. 

*Structure-related rights*. Structure-related rights allow parties to add and remove other parties as well as to edit their rights at any time. *Object-related rights*. Object-related rights allow added parties to perform the creation and deletion of objects and, in between, the execution of an arbitrary sequence of transactions, transformations, aggregations, and, in the case of previous aggregations, the execution of disaggregations. 

As stated in Section 4.2.1, after the dApp’s deployment, the deploying account becomes the initial administrator and, therefore, has full access to all functions. Subsequently, the administrator can edit the rights of all involved parties at any time. 

### 4.3. Development of a Token Concept

Identifiable and traceable objects are at the heart of traceability systems [2]. It, therefore, follows that a blockchain-based traceability system rests upon traceable asset tokens. Therefore, as an initial step, it requires the development of a token concept that provides a fundamental structure to allow the integration of all object-related supply chain events.

Existing dApp architectures, such as the approach from Westerkamp et al. [44], show that NFTs based on the ERC-721 demonstrate strong capabilities when identifying objects in supply chains. However, they have weaknesses when the minting of a token depends on the fulfilment of conditions containing other tokens. Westerkamp et al. [44] refer to such smart contracts containing minting conditions as ‘token recipes’. In supply chains, this weakness typically becomes evident when mapping batch productions that involve assembling parts. The nature of the problem is that it is impossible to predict tokens’ addresses; therefore, conditions of a token smart contract can only include required tokens subsequent to their minting. This issue is of no consequence for assemblies produced in lot size one. However, this problem considerably increases the mapping complexity for producing non-fungible assembly batches aggregating inputs of the same type since every assembly requires deploying an individual recipe contract. 

The ERC-1155 standard described by Radomski et al. [51] and adopted by the supply chain mapping architecture of Kuhn et al. [46] solves this problem to some extent; however, only for fungible batches aggregating fungible components. The ERC-1155 standard allows the minting of non-fungible token types, which allows the minting of a group of FTs of the same type (see Section 2). Therefore, the ERC-1155 standard sets the foundation for including token types as minting conditions in the token recipe when mapping tokens represent assemblies. Therefore, compared to the ERC-721 standard, the ERC-1155 standard simplifies the mapping complexity for fungible assembly batches aggregating fungible inputs of the same type since, in this scenario, *one* token recipe can mint *multiple* assembly tokens. However, when mapping multiple non-fungible assemblies of the same type with non-fungible inputs of the same type, the ERC-1155 reaches the same limitations as the ERC-721. 

Thus, no token standard and logic allows an efficient mapping of non-fungible assembly batches of the same type, aggregating non-fungible inputs of the same type. Figure 3 illustrates the required interplay of NFTs and minting conditions for assembly processes. As the example shows, it essentially requires a token concept that allows the minting of batches of NFTs of the same type with one smart contract. Like this, it is possible to include the inputs’ NFT types as minting conditions, and *one* token type recipe can mint *multiple* non-fungible assembly tokens of its type.

The architecture proposes adopting a new token concept that allows minting batches of NFTs of the same type to solve the batch production problem. Therefore, the architecture extends the idea behind token recipes containing the minting conditions for NFTs and integrates this functionality in non-fungible *blueprints* for NFT types. This paper defines the emerging token blueprints as follows:


*A blueprint defines the minting conditions for non-fungible tokens necessary to mint multiple non-fungible tokens of its token type. Like non-fungible tokens, each blueprint is unique and has an owner. However, unlike non-fungible tokens, blueprints cannot change their owner.*


Similar to ERC-721-based tokens, only the pair of smart contract addresses and blueprint IDs make the blueprint’s token type globally unique. The NFTs, on the other hand, require the combination of smart contract addresses, blueprint IDs, and token IDs for their global uniqueness. Therefore, with its ability to group NFTs into unique token types, the proposed blueprint-based token concept aims to overcome the limitations of the existing NFT standards, ERC-721 and ERC-1155. Figure 4 shows the positioning of the proposed blueprint-based tokens in the NFT standard landscape. 

#### 4.3.1. Integrating Object Events

As summarised in Section 1, object events consist of two sub-events: creation and deletion. When transferred to a blockchain-based architecture, this requires the functionality of minting tokens and their deletion. Therefore, on a system level, object events determine the beginning and the end of mapping objects throughout their life cycle and set the structural data frame for all object-related supply chain events mapped by the architecture.

Since every token stems from a blueprint, the creation of blueprints marks the precondition for minting tokens. Hence, the architecture requires the governance set library to be extended with an additional *blueprint set*. The blueprint set contains all blueprint-related functions and manages all existing blueprints in a database structure—the *blueprint memory*. Clients can create unique blueprints and define the characteristics of their represented object type. Since, unlike tokens, blueprints cannot change their owner, the initiator of a blueprint automatically becomes its owner and thus has access to its functions.

With available blueprints, accounts have the possibility to mint tokens. This requires the architecture to have a third library—the *token bucket*. The token bucket includes all token-related functions and, similar to the other libraries, stores and manages available tokens in a *token memory*. 

Fundamentally, as the architecture’s token concept indicates, tokens are the logical results of their blueprint and its minting conditions. Here, the architecture utilises a hashing mechanism to ensure the uniqueness of blueprints and tokens. Figure 5 illustrates the hashing mechanism used for minting NFTs of the same type. As indicated, blueprints are a logical result of its data input and the blockchain’s timestamp, while the token ID is a logical result of its blueprint and the blockchain’s timestamp. Consequently, one blueprint can create multiple NFTs of the same type, each having a unique timestamp and token ID.

In order to delete tokens, this paper’s architecture utilised the same mechanism for removing parties from the governance set. Therefore, the functionality of deleting tokens removes tokens from the token bucket, which updates the token memory and affects all blocks after the ‘removal-transaction’. However, the blockchain’s immutability ensures that deprecated blocks with a deprecated state of the token bucket still prove the tokens’ past existence. In addition, the blockchain’s metadata can always reveal the address that triggered the token deletion transaction.

#### 4.3.2. Integrating Aggregation/Disaggregation Events

While minting tokens represents object events without conditions, strictly speaking, on a system level, the aggregation of tokens represents object events under the fulfilment of object-related minting conditions. As described in the token recipes by Westerkamp et al. [44], object-related minting conditions define the input tokens for minting token aggregations. The immutability of blockchain technology ensures that it is impossible to create token aggregations without owning the input tokens specified in the minting conditions. 

In EPCIS [9], aggregation events result in a new entity containing the input objects. From this moment on, aggregated objects, physically and on a system level, occupy the same location at the same time. The architecture aims to map this exact situation in a token data structure, particularly to facilitate the traceable disaggregation of tokens in later stages. Therefore, while the previous approaches by Westerkamp et al. [44], Watanabe et al. [45], and Kuhn et al. [46] ‘consume’ or ‘burn’ the required tokens when aggregating them into a new token, this architecture proposes an alternative aggregation mechanism.

For this mechanism, the architecture extends the token bucket with another memory—the token container. Instead of consuming or burning tokens, triggering an aggregation initiates pushing the required tokens from an account’s token memory to the token container before creating the aggregated token and placing it in the account’s token memory. Subsequently, the emerging aggregated token ID references its containing tokens. 

Figure 6 illustrates the required request and transaction flow for aggregating tokens in the token bucket. As indicated, before aggregating tokens, the request flow must first clarify an account’s access rights specified in the governance set, the availability of the blueprint in the blueprint set, and the availability of all required tokens in the token bucket. Subsequently, the account can trigger the transaction and add the aggregated token according to the described aggregation mechanism.

When disaggregating previously aggregated tokens, this paper’s architecture proposes a mechanism that restores previously aggregated tokens and thus ensures consistent traceability of children tokens. An initiated disaggregation deletes the parent token from the token memory, which breaks the reference to its children token. This, in turn, pushes the children tokens from the token container back to the token memory. For the deletion of the parent token, this architecture integrates the same deleting mechanism described in Section 4.3.1 in the disaggregation function. 

Figure 7 summarises the mechanism of the token bucket to aggregate and disaggregate tokens. As illustrated, the aggregated tokens remain in the token container. However, they are not accessible in the token memory to experience individual events and instead become a fixed component of their superordinate aggregated token and its experienced events. Therefore, in a simplified way, aggregated tokens in the token container refer to their parent token as their owner instead of a public key. The disaggregating mechanism simply reverses this process and restores the previously aggregated tokens.

#### 4.3.3. Integrating Transformation Events

Characteristically, data related to an object’s type remains static, while the individual objects themselves can transform when experiencing state changes, for example, regarding their quality, approvals, or processing [46]. Unlike aggregations, which result in a new token with a new identifier, transformations maintain the same identifier and only update the token-related metadata [45]. Transferring this general structure of token transformations to the paper’s architecture requires a mechanism to update the tokens’ metadata. The possibility for each token to experience transformation events also demonstrates the requirement of the paper’s token concept that each token has individual metadata and not merely a copy of the blueprint’s metadata, usually representing static data describing the token type. 

At first, in its initial state (state i), a token receives the metadata defined when minting it. Afterwards, in state i+1, the token experiences a transformation resulting in a change regarding its metadata. From this moment on, the current state of the token memory always links the token ID to the updated metadata. Similar to the party or token deletion mechanism, only the latest blocks reflect the tokens’ current metadata. However, deprecated blocks still show tokens’ past metadata. 

#### 4.3.4. Integrating Transaction Events

Committing transactions of tokens is an integral element of every token standard. Therefore, this architecture adopts the established transfer function of the existing NFT standards ERC-721 and ERC-1155 [50,51] and integrates it into the token bucket. As described in Section 4.3.1, when minting a token, the account triggering the transaction becomes the first assigned owner of the token. The same applies to tokens resulting from a token aggregation, as described in Section 4.3.2. Transaction events allow token owners to transfer tokens from their account to another, permitting the new owner to execute all types of supply chain events. Since the token memory of the token bucket stores tokens in a database structure, a token transfer represents an update of the owner column. This results in a similar mechanism to transforming tokens, except that the transfer transaction updates the tokens’ owner instead of the metadata.

#### 4.3.5. Integrating a Token History 

“Although the existing token standards focus on secure input and interface design, they do not consider an efficient way of conducting history searches” [45]. Therefore, it requires searching the blockchain’s metadata to acquire visibility regarding tokens’ histories, typically by means of a blockchain transaction explorer. Consequently, the traceability complexity can increase drastically depending on the lengths of tokens’ life cycles on the blockchain and the entanglement of their experienced object-related event sequences [45]. This contrasts traceability systems’ actual objective and requirements to ensure objects’ composition and processing traceability [2]. 

To overcome these traceability challenges, Watanabe et al. [45] propose extending tokens with a ‘pointer’ referring to their previously experienced object event to facilitate the accessibility of tokens’ histories. Kuhn et al. [46], on the other hand, propose an additional dApp designed explicitly for reconstructing tokens’ histories in a smart contract. The present paper’s architecture combines these two approaches and proposes making the tokens’ history an integral part of the token bucket. In this way, minting a token generates a copy of the token in an additional database structure—the token history. In the token history, each object-related event subsequent to the token minting points to the previously experienced event. Therefore, minting a token merely sets the starting reference of the token history and, without other object-related events, represents the only data entry in the array. In this context, the illustration of aggregation and disaggregation events requires alternative forms of presentation. 

In particular, to map the history of children tokens after a disaggregation event, the token history requires a new logic to ensure their traceability since the parent token is no longer a reference point for further events. Therefore, this paper’s architecture proposes generating a ‘disaggregation object’ in the token history which references the children tokens after their disaggregation. This ensures that whenever the history of a token references a disaggregation object, the child token is not part of an aggregation anymore and can again experience an individual event history. Accordingly, future events reference the previous child token instead of the disaggregation event. Figure 8 shows an example of the token history reference mechanism when aggregating and disaggregating tokens.

## 5. Prototyping-Based Evaluation 

A recent comparison between dApp-capable blockchain platforms shows that the Ethereum blockchain represents the most advanced blockchain platform, responsible for 84.5% of all deployed dApps across the platform candidates [52]. Accordingly, this paper uses the Ethereum blockchain and the Ethereum-specific development tools and environments for the dApp architecture’s prototypical implementation. Therefore, the architecture prototype uses Truffle Suite (v5.5.13) as the development environment for the dApp and deploys the smart contracts on a local Ganache Ethereum Blockchain (v7.1.0). As common for Ethereum dApps, the smart contracts use the Solidity programming language with the compiler Solc-JS (v.0.8.8) and NodeJS (v15.14.0) as the JavaScript runtime environment for Truffle Suite. For the interaction with the local Ganache Ethereum blockchain and the smart contract, the prototypical dApp uses the Ethereum JavaScript API Web3JS (v1.3.5) and MetaMask as the wallet provider. For the frontend development, the prototypical dApp uses ReactJS (v17.0.2) and TailwindCSS (v3.1.8) as the user interface design framework. 

Currently, the Ethereum blockchain limits the smart contract size to 24,576 bytes [56]. Due to the complexity of the dApp’s architecture and the number of required functions, this limit prevents the implementation of the complete logic in one smart contract. Alternatively, the Ethereum blockchain allows smart contracts to be split into several libraries, outsourcing certain functions. When applied to the proposed dApp architecture, this requires outsourcing the three components’ governance set, blueprint set, and token bucket, each in its own smart contract. The supply chain smart contract serves as a connector for all three libraries, maintaining the structure of the original architecture. 

In order to evaluate the architecture components’ practical applicability holistically, the evaluation uses an example of a supply chain. This describes a simple process where a manufacturer receives a delivery from a transport company, which includes a component from its supplier. Subsequently, the manufacturer unpacks the delivery, deletes the delivery box, and processes the component from the supplier. Finally, the manufacturer assembles the processed component along with one of their in-house components. Figure 9 illustrates the example supply chain utilised to demonstrate the dApp’s workflow.

Before starting the process, the dApp administrator must register all parties involved in the dApp. In the supply chain example case, the manufacturer deploys the smart contracts and, therefore, has the initial rights to add further parties. As indicated, the interface aims to deanonymise a party’s public key and assign initial rights to it. To promote clarity, each party receives a role in the supply chain. The administrator can add new roles and assign them to the party if a role does not exist. 

Listing 1 shows the source code for adding parties. Fundamentally this function allows a party with the rights to add other parties to insert a public key, the *partyAddress*, and deanonymise it by adding information such as the name, contact data, and role. In addition, the added party receives an assignment of initial rights. Here, as proposed by the architecture, the prototype differs between administrative structure-related rights and operative object-related rights.

**Listing 1.** Source code for adding parties.1 function addParty(2  string memory partyName,3  string memory partyContact,4  string memory roleName,5  string memory roleColor,6  address partyAddress,7  bool operativeRights,8  bool administrativeRights9 ) public onlyPartiesWithAdministrativeRights {10  Party memory party = Party(11   partyAddress,12   partyName,13   partyContact,14   roleName,15   roleColor,16   basicCreationRights,17   canAddParty18  );19  parties.insertParty(party);20 }

The next step requires all added parties to create new blueprints of their respective parts. Here, the assembly creation requires component 1 and component A ownership, and the blueprint defines these two components as minting conditions. All existing blueprints are globally visible to all registered accounts when defining the minting conditions. Here, the unique blueprint ID of each available blueprint ensures the exact conclusion of the correct blueprints in the requirements. For example, including the blueprint of component 1 translates to the requirement of owning a token created with the component 1 blueprint.

The function of creating blueprints represents the key element of the blueprint set. Listing 2 shows the respective source code. For the unique blueprint ID creation, the function hashes the token type name, its metadata, and the creation timestamp. As a hashing algorithm, the function uses the Ethereum typical keccak-256. Like the SHA-256, the keccak-256 results in 64 bytes in a hexadecimal number. In addition, the function allows the inclusion of other blueprints as minting conditions. 

**Listing 2.** Source code for creating blueprints.1 function createBlueprint(2  BlueprintItSet storage self,3  address key,4  string memory token,5  string memory description,6  string memory partyName7 ) public returns (bytes32, Blueprint memory) {8  uint256 creationTimestamp = block.timestamp;9  bytes32 blueprintId = keccak256(10   abi.encode(token, description, creationTimestamp)11  );12  Blueprint memory createdBlueprint = Blueprint(13   blueprintId,14   token,15   description,16   supplierName,17   creationTimestamp18  );19  self.supplierBlueprints[key].push(createdBlueprint);20  self.allBlueprints.push(createdBlueprint);21  return (blueprintId, createdBlueprint);22 }

In order to mint tokens and perform token aggregations, all parties require to access the *createToken* function. Listing 3 shows the source code of the createToken function. This function represents an essential element of the token bucket and the dApp in general. It allows the minting of new tokens and includes the possibility of minting tokens with minting conditions. Therefore, the function summarises the minting and aggregation of tokens in one logic. As the logic of the architecture suggests, the creation of tokens requires the respective blueprints. In case of aggregation, the function pushes the aggregated tokens into the token container of the token bucket, creates a new token ID, and deletes the aggregated tokens from the token memory. In addition, the function adds the creation data to the token history. 

**Listing 3.** Source code for creating tokens.1 function createToken(2  TokenMap storage self,3  address key,4  Blueprint memory blueprintData,5  uint256 creationTimestamp,6  bytes32 tokenId,7  string memory tokenDescription,8  Token[] memory tokensToMerge9 ) public {10  require(bucketContainTokens(self, tokensToMerge));11  bytes32 blueprintId = blueprintData.blueprintId;12  bytes32[] memory aggregatedTokensIds = new bytes32[](13   tokensToMerge.length14  );15  string memory tokenName = blueprintData.token;16  Token memory token = Token({17   tokenId: tokenId,18   blueprintId: blueprintId,19   tokenName: tokenName,20   tokenDescription: tokenDescription,21   tokenCreator: blueprintData.blueprintCreator,22   creationTimestamp: creationTimestamp23  });24  for (uint256 i = 0; i < tokensToMerge.length; i++) {25   self.containedTokens[tokenId].push(tokensToMerge[i]);26   aggregatedTokensIds[i] = tokensToMerge[i].tokenId;27   deleteToken(self, key, tokensToMerge[i].tokenId, true);28  }29  insertToken(self, key, token);30  insertHistoryData(31   self,32   tokenId,33   tokenName,34   tokenDescription,35   msg.sender,36   msg.sender,37   creationTimestamp,38   aggregatedTokensIds,39   TokenHistoryState.Creation40  );41 }

In the example supply chain, the manufacturer must disaggregate component 1 from the delivery box. The function to disaggregate tokens is another key element of the architecture since it represents a major distinction compared to previous architectures. Listing 4 shows the source code for disaggregating tokens. As the functions indicate, the disaggregation reverses the aggregation processes and moves the previously aggregated tokens from the token container back to the token memory and, therefore, back to the inventory of the triggering account. Subsequently, the function calls the *deleteToken* function in the token bucket and deletes the aggregation token from the token memory. Finally, the function adds a disaggregation object to the disaggregated tokens in the token history.

**Listing 4.** Source code for disaggregating tokens.1 function disaggregateToken(2  TokenMap storage self,3  Token memory token,4  Token[] memory tokens,5  address key6 ) public {7  bytes32 tokenId = token.tokenId;8  require(contains(self, key, tokenId));9  uint256 disaggregationTimestamp = block.timestamp;10  bytes32[] memory disaggregatedTokenIds = new bytes32[](tokens.length);11  for (uint256 i = 0; i < tokens.length; i++) {12   Token memory disaggregatedToken = tokens[i];13   disaggregatedTokenIds[i] = disaggregatedToken.tokenId;14   insertToken(self, key, disaggregatedToken);15  }16  deleteToken(self, key, tokenId, false);17  insertHistoryData(18   self,19   tokenId,20   token.tokenName,21   token.tokenDescription,22   msg.sender,23   msg.sender,24   disaggregationTimestamp,25   disaggregatedTokenIds,26   TokenHistoryState.Disaggregation27  );28 }

The manufacturer aggregated the transformed component 1 with component A in the final step. Since the architecture proposes storing the token composition and history on-chain, the data structure allows accessing each token’s entire composition and history directly via the interface.

Figure 10 shows the respective interface for accessing the token history. The interface displays the immutable history of the example of the event flow of component 1. As illustrated, this includes the exact timestamps of all experienced events in chronological order, as well as the parties involved. Here, the interface reduces the blockchain-typical designations, such as public key strings, and uses the registered parties’ names to promote comprehensibility. A key element of the token history is that even though component 1 experienced an aggregation and a disaggregation, as the token IDs indicate (0x98d…), the transformation subsequent to the disaggregation happens to the same token as before the aggregation. This not only facilitates the traceability of complex supply chain event flows but also highlights the architecture’s capability to restore tokens when conducting disaggregations.

## 6. Results

This paper proposes a blockchain-based traceability architecture that integrates a governance concept and a novel token concept to overcome the limitations of the existing architectures. Here, the governance concept serves to deanonymise parties and manage the supply chain structure by adding and removing parties as well as editing their application rights. The novel token concept introduces token ‘blueprints’, which allow clients to mint multiple tokens of different token types, where each token of the same type is non-fungible. Furthermore, blueprints can define minting conditions necessary to mint token aggregations that require the ownership of components, for example, when mapping assemblies. This structure facilitates the integration of functions in the architecture, which allows for conducting all object-related supply chain events: creations and deletions, aggregations and disaggregations, transformations, and transactions. In addition, the token concept contains logic for an integrated token history, which ensures accessible on-chain traceability of all object-related events and, therefore, the end-to-end traceability of their physical or abstract representatives. Therefore, compared to available advanced blockchain-based traceability architectures, the proposed architecture enables a holistic coverage of the object-related supply chain events defined by EPCIS and incorporates a governance concept as an integral component of the dApp. Table 1 compares the available advanced blockchain-based traceability architectures with the architecture proposed in this article.

As the prototyping-based evaluation shows, Ethereum’s programming language, Solidity, enables the practical applicability of all developed components. Furthermore, the Web3 user interface connected to the MetaMask wallet allows access to all smart contracts’ functions. By means of a supply chain example, it is possible to visually demonstrate the functioning of novel components, such as token blueprints and the integrated token history. However, it is necessary to outsource functions in smart contract libraries to prove not only the practical applicability of individual components but also to combine them holistically in a complete blockchain-based traceability solution. Therefore, due to Ethereum’s current smart contract size limitation of 24576 bytes, the prototype splits the functions of the governance set, blueprint set, and token bucket, each into its own smart contract library. A central supply chain smart contract connects all three libraries and maintains the architecture’s original structure.

The necessity to split smart contracts into libraries enforces a complexity dilemma for dApps with extensive relationships and functional dependencies caused by the quasi-Turing completeness of Ethereum dApps. The quasi-Turing completeness of Ethereum dApps results from the fact that the underlying blockchain’s transaction size limits the number of computational steps for executing the smart contracts machine [47]. Consequently, the transactional input can exceed not only the possible transaction size but also the computational steps caused by transactional queries. Figure 11 illustrates the dApp complexity dilemma.

As Figure 11 illustrates, the number of transactional queries and the transactional input size both increase the transaction size and limit the possible dApp complexity. Since the architecture’s complex update and request relationship eventually increase the required transactional queries, the prototype in its current form already exploits Ethereum’s possible dApp complexity, even though the actual transaction inputs in the example supply chain are relatively small. For example, adding a more granular rights allocation for each function—instead of generalising the rights to administrative structure-related and object-related operative rights—inevitably increases the functional dependencies and, therefore, the required transactional queries, which exceeds Ethereum’s current transaction size limit.

## 7. Discussion

This section summarises the paper’s key findings by answering the defined research questions.

*PRQ* 
*How can a blockchain-based traceability architecture be constructed which meets the general-purpose requirements of dynamic, interconnected supply networks and ensures end-to-end traceability of object-related supply chain events?*


The paper’s architecture applies the fundamental structure of Web3 applications: interfaces, smart contracts, and an underlying blockchain platform serving as the operating system. It is composed of its smart contract structure’s three components; the governance set, blueprint set, and token bucket. Here, the governance set includes functions defining the supply chain structure-related administrative capabilities of the dApp architecture. Therefore, compared to previous approaches, which typically rely on permissioned blockchain settings to ensure dApp governance, the proposed architecture’s governance concept is an integral component of the dApp. The functions of the governance set include the possibility to add and remove parties as well as to edit their structure-related administrative and object-related operative rights.

Furthermore, the architecture entails a blueprint set, allowing clients to create ownable and unique token blueprints, thus introducing a novel concept that defines minting conditions for NFTs necessary to mint multiple NFTs of the same token type. In this way, the novel concept of blueprints solves the limitations of the current ERC-1155 token standard, which only allows minting multiple FTs of the same token type.

The token bucket, which defines the object-related operative capabilities of the dApp, represents the architecture’s final component. The token bucket contains functions allowing the creation and deletion, aggregating and disaggregating, transforming, and transferring of tokens. Therefore, compared to the previous architectures, the proposed architecture covers all object-related supply chain events defined by the EPCIS standard. Furthermore, since previous architectures cannot directly perform token disaggregations, the proposed architecture establishes a sub-component of the token bucket, the token container, which stores all tokens that are part of token aggregations. Instead of referencing an owning account, tokens in the token container reference their aggregation token. When disaggregating tokens, the logic deletes the token aggregation and pushes its containing tokens back to the account’s inventory. This ensures a mechanism allowing the restoration of previously aggregated tokens. Finally, the token bucket has an integrated token history, which ensures accessible on-chain traceability data of all object-related events and, therefore, the end-to-end traceability of their physical or abstract representatives.

*SRQ1* 
*What are the limitations of existing blockchain-based traceability solutions described in the literature?*


The vast majority of existing solutions deal with low-complexity architectures allowing the traceability of single objects without the ability to map compositional changes. However, the three advanced blockchain-based traceability architectures developed by Westerkamp et al. [44], Watanabe et al. [45], and Kuhn et al. [46] show certain general-purpose capabilities that can map compositional changes. First, the Westerkamp architecture applies the ERC-721 NFT standard as an architectural means and introduces ‘token recipes’ to map token aggregations. The Watanabe architecture applies the same ERC-721-based logic and extends tokens with ‘pointers’ to each token to the state of the past tokens to facilitate traceability. Lastly, the Kuhn architecture applies the ERC-1155 hybrid token standard and incorporates a governance concept based on a permissioned Ethereum blockchain. However, the applied architectural means show limitations regarding their ability to map all object-related supply chain events and an integrated governance concept. The limitations regarding object-related supply chain events specifically involve the deletion of tokens and aggregation and disaggregation.

*SRQ2* 
*What are the architectural requirements for an end-to-end traceability solution for dynamic, interconnected supply networks?*


As typical for traceability systems, dynamic, interconnected networks also require parties and objects at their heart. The architecture must clearly identify each party and assign role-related rights. Furthermore, since interconnected supply chains are subject to structural transitions at any time, the architecture must ensure that parties are integrated seamlessly but must also have the capability to remove them if necessary. In addition to the structural requirements, interconnected supply chains place requirements regarding the objects. These require the clear identification of physical and abstract objects as well as the capability to map all object-related supply chain events, which involves their creation and deletion and, in between, the execution of transactions, transformations, aggregations, and, in the case of previous aggregations, the execution of disaggregations. Due to the emergent characteristics of contemporary interconnected networks, objects travel in a non-predefined manner through supply chains which requires the architecture to allow objects to experience events in an arbitrary sequence. Finally, to ensure the architecture’s traceability feature, the architecture requires the provision of a traceable end-to-end event history throughout an object’s life cycle.

*SRQ3* 
*How can the architecture’s components be implemented in code to enable its practical applicability in a blockchain-based traceability solution?*


The prototyping-based evaluation proves the practical applicability of all architectural components by utilising a local Ethereum platform with Ethereum-specific development tools and environments. Furthermore, it is possible to implement the component’s update and request relationships in a blockchain-based traceability solution. However, due to Ethereum’s current smart contracts’ size limitation of 24576 bytes, it is necessary to outsource functions in smart contract libraries. Here, the prototype uses libraries according to the architecture’s structure and establishes libraries for the governance set, the blueprint set, and the token bucket. Finally, a central supply chain smart contract maintains the architecture’s original logic and queries the functions outsourced in the respective libraries.

Since the quasi-Turing completeness of the underlying blockchain limits the number of computational steps for executing smart contracts, smart contract functions that contain functional dependencies, such as calling other functions or requesting data, increase the size of transactions from a computative standpoint. Therefore, not only can the transactional input exceed the possible transaction size, but also the computational steps caused by transactional queries. Consequently, the necessity to outsource functions in smart contract libraries enforces a complexity dilemma for dApps with extensive relationships and functional dependencies, such as the proposed blockchain-based traceability solution. The complexity dilemma describes a dilemma where the number of transactional queries and the transactional input size both increase the transaction size and limit the possible dApp complexity. Even though the transactional input size of all transactions conducted within the prototype-based evaluation is relatively small, the architecture’s functional dependencies and relationships eventually require querying data, such as account rights or blueprint and token availability. This ultimately increases the necessary transactional queries and thus increases the computational steps for executing smart contracts. Consequently, the prototype in its present state exploits the currently possible dApp complexity of Ethereum dApps.

## 8. Summary of Unique Contributions

This section summarises and lists the paper’s unique research contributions resulting from answering the research questions.

*Integrated governance concept*. Compared to previous advanced blockchain-based traceability architectures, the proposed architecture incorporates a governance concept as an integral component of the dApp. The governance concept includes all functions necessary to manage the supply chain’s structure. This makes the dApp independent of the underlying blockchain access settings since it allows an operation in both permissionless and permissioned settings. Thus, the blockchain platform serves solely as an operating system determining the framework conditions for smart contracts and smart contract interactions but does not actively need, for example, to deanonymise parties and allow their participation.*Blueprint-based token concept*. The proposed blueprint-based token concept introduces a novel NFT logic that overcomes the limitations of available ERC-721 and ERC-1155 token standards, which previous supply chain traceability architectures commonly applied for reflecting supply chain objects. Compared to available token standards, the introduced blueprints enable minting multiple NFTs of the same type. This facilitates mapping token aggregations since minting conditions can include the type instead of a specific token ID. Therefore, instead of requiring a smart contract recipe for each assembly, one blueprint can mint multiple non-fungible assembly tokens of its type.*Coverage of object-related supply chain events*. The proposed blockchain-based traceability architecture covers all supply chain events defined by EPCIS. In particular, the proposed architecture includes new mechanisms to aggregate and disaggregate tokens. For this to be possible, the architecture pushes aggregated tokens into an owned memory, the ‘token container’. Simplified, instead of referring to an owning public key, aggregated tokens refer to an ‘owning’ token aggregation. The disaggregation function pushes the previously aggregated tokens from the token container back to the token memory and deletes the token aggregation. This mechanism ensures that previously aggregated tokens are restored and maintain their unique identifiers, representing a cornerstone for mapping sequences of aggregations and disaggregation, as is necessary for delivery supply chains.*Prototypical implementation*. The paper provides a fully operational prototype that covers all functionalities specified in the blockchain-based traceability architecture. However, due to the architecture’s complexity and Ethereum’s current transaction limit, the prototypical implementation reveals a dApp complexity dilemma. This describes, for the first time, that complex dApps requiring extensive functional dependencies not only experience limitations regarding their transactional inputs (for example, storing images on-chain) but also face limitations regarding their possible computational steps when executing transactional queries.

## 9. Conclusions and Recommendations

The blockchain-based traceability architecture developed in this paper provides an environment that reflects objects with globally unique tokens without requiring a central coordinator. At the same time, the proposed architecture shows completeness regarding the mapping of object-related supply chain events defined by EPCIS and proves general-purpose capabilities applicable to manufacturing and logistics scenarios. This, in particular, includes new mechanisms to aggregate and disaggregate tokens. Here, blockchain’s immutability ensures data consistency across all participating parties, for example, when defining aggregation conditions for mapping assembly processes, resulting in the ability to ensure consistent reflections of tokens’ histories of their arbitrarily experienced sequences of supply chain events. These elements represent important cornerstones in making blockchain’s end-to-end traceability capabilities accessible for complex supply chains containing objects that can experience compositional changes. Thus, the architecture overcomes the shortcomings of current blockchain-based traceability solutions. However, specific scenarios may require an extension of its capabilities. Therefore, as the final step, the paper summarises the architecture’s current limitations and anticipates topics for further research.

*Blockchain scalability*. The paper’s dApp architecture proves the technical feasibility of a blockchain-based architecture allowing a holistic mapping of object-related core events. This serves as an example of blockchain’s strong and incomparable capabilities regarding objects’ traceability in dynamic, interconnected supply networks. However, the architecture’s prototypical implementation already exploits the currently possible dApp complexity of permissionless environments in a supply chain management context. Most certainly, further research is necessary to drastically improve blockchain technology’s scalability to provide blockchain-based operating systems with the capacity to deploy dApps creating traceability ecosystems with industrial relevance. This includes all blockchain’s major elements, such as block size, transaction fees, transaction size, transactional throughput, and smart contract size.*Complexity dilemma*. The prototypical implementation focuses on proving the architecture’s practical applicability. Consequently, the prototype exploits Ethereum’s possible dApp complexity in its current version. Extending the possible dApp complexity correlates strongly with the previously stated limitation regarding improving blockchain’s general scalability. However, optimising the architecture’s components in terms of algorithm efficiency and reducing functional dependencies may lower its required computational resources. Further research is necessary to evaluate the proposed concept mathematically and identify resource-intensive inefficiencies.*Legal possession*. The architecture displays only the current owner of objects but not their legal possession. In cases where the legal owner is crucial for the application scenario (e.g., in leasing business models), an extension may be necessary. Here, one solution could be to extend the wallet functionality and introduce two types of token transactions, changes of ownership and legal possession. In general, blockchain tokens and their transparent and immutable traceability cannot currently replace legally binding documents since legal authorities do not yet accept the technology as a legal basis. Further research may investigate the capabilities of blockchain technology regarding its suitability for objects’ legally-binding proof of possession.*Standardised interfaces for objects*. The architecture allows adding metadata as individual input data strings and, therefore, generalises the inputs for all objects, making it vulnerable to incorrect input and inconsistent data formats. Since research regarding cumulative approaches such as the AAS aims to create standardised interfaces and data formats for each object type, it may be possible to combine the two approaches efficiently. A possible combined solution could rely on the blockchain-based dApp architecture as a superordinated data structure and attach object-related data using the standardised interfaces of the AAS.*Object durability*. The architecture assumes that even after several transformations, a disaggregation of an assembly into its components is always possible. However, there may be scenarios where transformations ‘destroy’ certain involved components. Currently, the architecture can map such cases only by stringing disaggregation and deletion events together. This requires a further investigation of the core object-related supply chain events and a possible division into further sub-events.*On-chain/off-chain storage*. The proposed architecture relies heavily on on-chain storage since it represents the most efficient way to retrieve interrelated information strings, such as the token history from the blockchain. However, such on-chain storage designs burden the blockchain, which already has a limited capacity regarding its block, transaction, and smart contract size. Therefore, further research is necessary to evaluate the architecture’s components regarding their data load in industrial settings over time, potentially requiring rearranging certain components in off-chain storages. This offers another intersection with the research regarding centralised cumulative approaches such as the AAS, possibly resulting in a hybrid architecture that utilises the AAS as standardised off-chain storage for sensitive and memory-intense data, while the blockchain stores data with high consistency necessity on-chain, utilising the logic of the proposed architecture.*Authority bubbles*. The architecture’s authority concept applies globally to the whole dApp. However, when objects leave one company’s ecosystem, it may be possible that these objects enter a new ecosystem with new roles and permissions. This raises the potential for exploring integrated ‘authority bubbles’ with a demarcated sphere of influence in a decentralised ecosystem.*Case study evaluation*. To address the previously mentioned limitations, it is necessary to further evaluate the architecture with industrial case studies located in different industries and domains, including emerging supply chain objectives in the sustainability domain, such as circular economy approaches and the generation of objects’ carbon footprints.

## Figures and Tables

**Figure 1 sensors-23-01410-f001:**
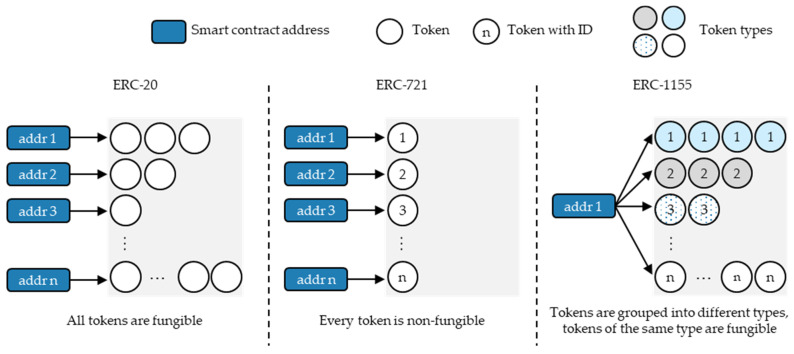
Overview of Ethereum token standards and their functionality (adapted from [48]).

**Figure 2 sensors-23-01410-f002:**
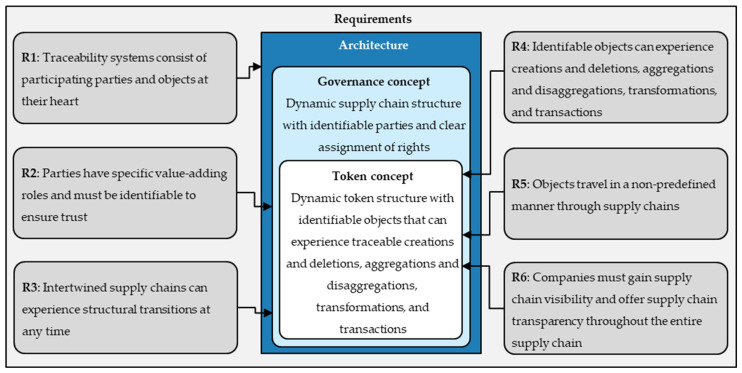
Architectural requirements and initial design suggestion.

**Figure 3 sensors-23-01410-f003:**
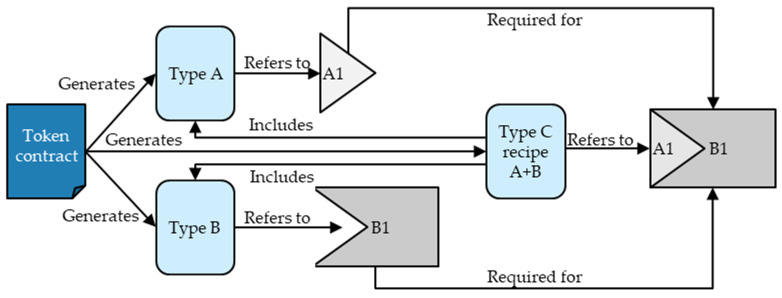
Required interplay of NFTs and minting conditions for assembly processes.

**Figure 4 sensors-23-01410-f004:**
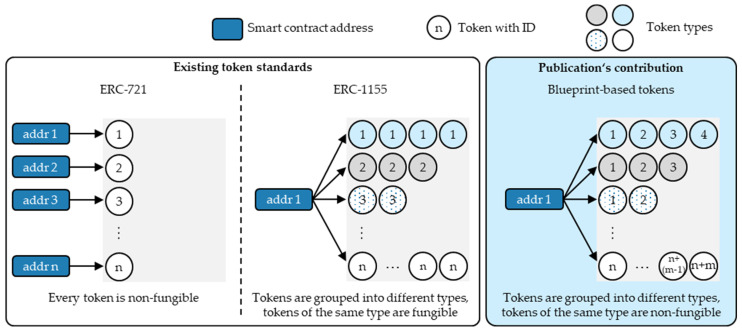
Positioning of blueprint-based tokens in the NFT standard landscape.

**Figure 5 sensors-23-01410-f005:**
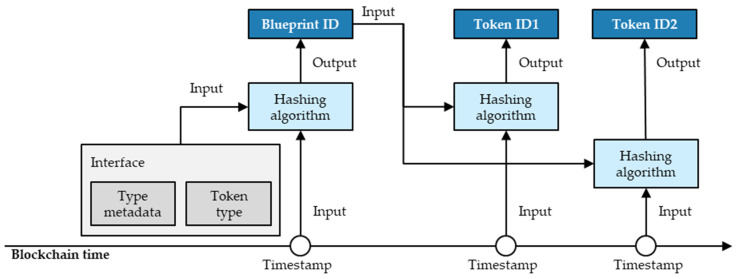
Hashing mechanism for creating NFTs of the same type.

**Figure 6 sensors-23-01410-f006:**
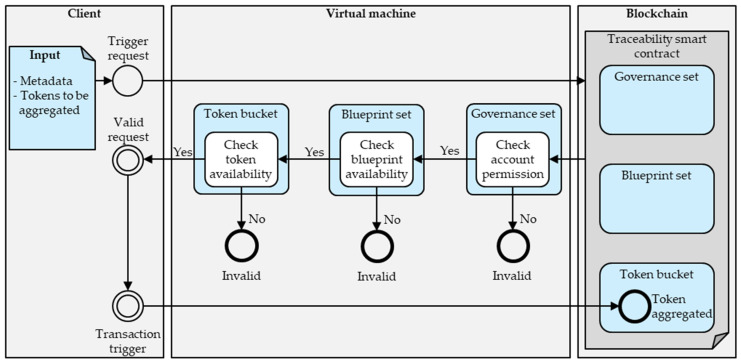
Request and transaction flow for aggregating tokens.

**Figure 7 sensors-23-01410-f007:**
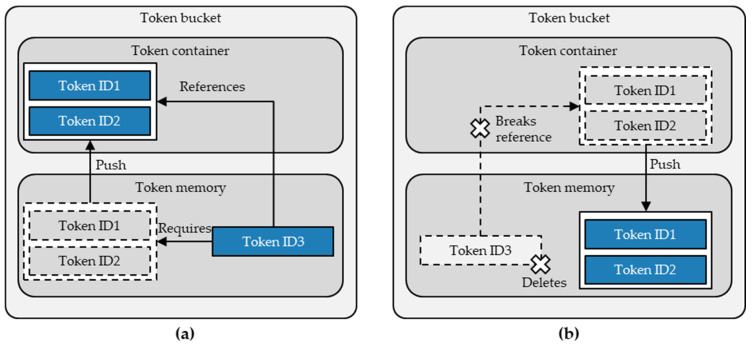
(**a**) Token aggregation mechanism; (**b**) Token disaggregation mechanism.

**Figure 8 sensors-23-01410-f008:**
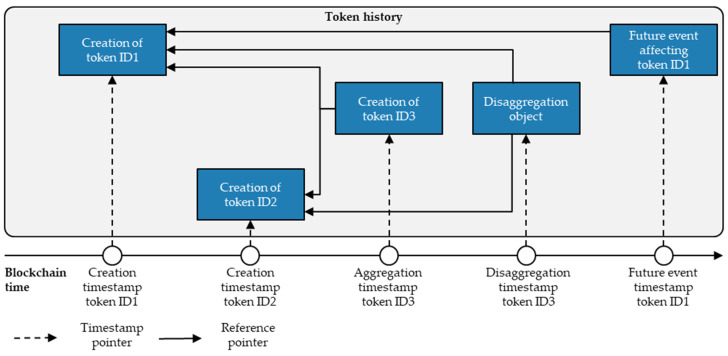
Token event history reference mechanism.

**Figure 9 sensors-23-01410-f009:**
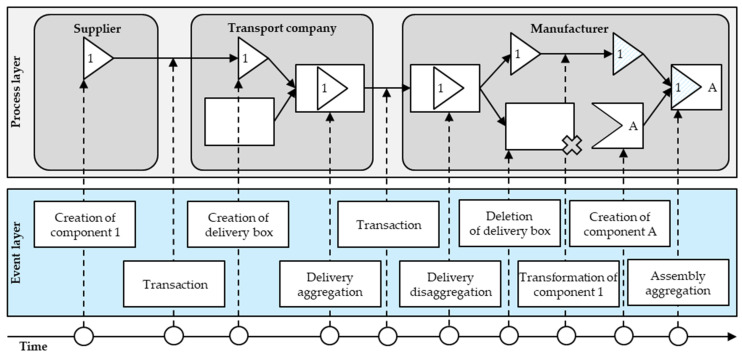
Example of a supply chain used to demonstrate the dApp’s workflow.

**Figure 10 sensors-23-01410-f010:**
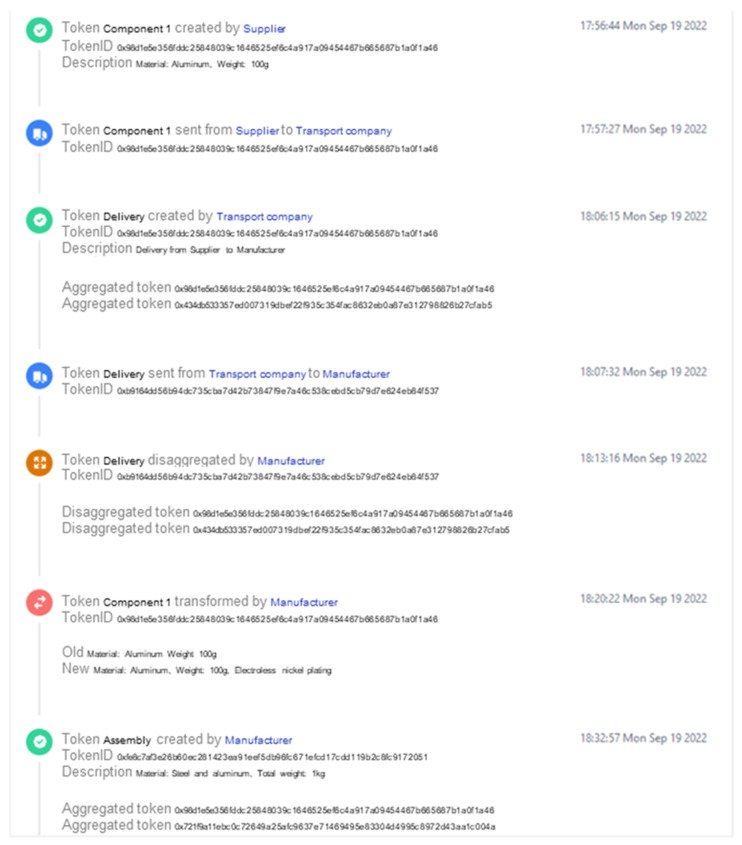
Interface for accessing the token history.

**Figure 11 sensors-23-01410-f011:**
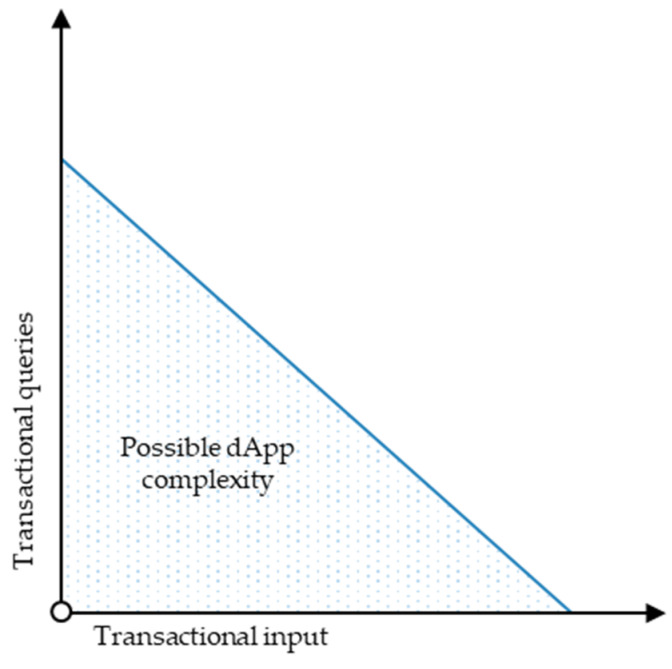
The dApp complexity dilemma.

**Table 1 sensors-23-01410-t001:** Comparison of advanced blockchain-based traceability architectures.

Architecture	Object Event		Aggregation Event		Transfor-mation Events	Transaction Events	Governance Concept	Token Concept
	Create	Delete	Aggregate	Disaggregate				
Westerkamp et al. [44]	x		x		x	x		ERC-721
Watanabe et al. [45]	x		x		x	x		ERC-721
Kuhn et al. [46]	x		x		x	x	x	ERC-1155
Article’s contribution	x	x	x	x	x	x	x	Blueprint-based

## Data Availability

No new data were created or analysed in this study. Data sharing is not applicable to this article.

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
