# Peer review of "Blockchain-Based Traceability Architecture for Mapping Object-Related Supply Chain Events"

_sensors, 2023, doi:10.3390/s23031410_

Round 1

Reviewer 1 Report

Authors' contribution is limited but can be elaborated with flow charts and step step algorithms. More work need to be done on writing style

Work can be accepted with a few modifications as suggested 

Reviewer 2 Report

The authors proposed a blockchain-based traceability architecture for supply chain events that is very latest and interesting idea. But in its current form manuscript have some weak points.

Abstract must be clear and concise concentrating at the aim of the research.

Mathematical aspect is very limited. the concept proposed here must be mathematically supported.

Results section must be extended by comparing with the similar schemes

Reviewer 3 Report

Dieterich et al. proposed a novel blockchain-based traceability architecture that integrates governance and token concepts to overcome the limitations of existing architectures regarding their object-related event mapping ability. The article is properly designed and structured. Results are clearly presented, and conclusions are well supported. This article can be improved if some of the following issues are addressed.

·         Scholars have been discussing the potential of blockchain and its applications in the supply chain sector. Indeed, blockchain can be considered an alternative solution to address some current issues. However, “blockchain can be applied” does not equal “blockchain should be applied”. What are the main issues faced by the supply chain sector? How many of them can be addressed by blockchain? Why do you think blockchain is THE solution compared to other technology-enabled solutions? Chapter 2 can be further strengthened.

·         Benefits introduced by the blockchain should be explained in more detail in Chapter 7. Can some quantitative analysis results be included here to strengthen conclusions? 

Reviewer 4 Report

This study proposes a blockchain-based traceability architecture that integrates governance and token concepts to overcome the limitations in the literature. I found the result of this paper interesting but very important issues have not been addressed. Therefore, I recommend a possible publication of the paper after a major revision. In the following, I report my specific comments.

- In the introduction, state the purpose of the work in the form of the hypothesis, question, or research problem, briefly explaining your rationale, and methodological approach, highlighting the potential outcomes your study can reveal, and describing the remaining structure of the paper.

- The last paragraph in section 3 should be developed and moved as a last paragraph of introduction.

- In the introduction part, the research method (DSR) adopted by the article is not well described or explained, which will make the readers confused.

- The authors focused on the research of blockchain-based traceability architecture but did not explain the technological background behind blockchain-based technologies in detail. What are the fundamental technologies needed to support the blockchain-based traceability architecture in supply chains?

- The conclusion part is also needed to be revised; which questions are answered, what is the contribution of the paper, and how the presented architecture answers the research questions that previous architectures are not able to answer?

- In the research methodology section, explanations are weak. The DSR method is a broad approach and more attention needs to be paid to its details. It is suggested to review the following paper:

--https://doi.org/10.25300/MISQ/2013/37.2.01

Round 2

Reviewer 4 Report

The authors have addressed the comments.